# Long-Term Follow-Up after Mycobacterium Chimaera Infection Following Cardiac Surgery: Single-Center Experience

**DOI:** 10.3390/jcm12030948

**Published:** 2023-01-26

**Authors:** Thibault Schaeffer, Sabine Kuster, Luca Koechlin, Nina Khanna, Friedrich S. Eckstein, Oliver Reuthebuch

**Affiliations:** 1Department of Cardiac Surgery, University Hospital Basel, 4031 Basel, Switzerland; 2Division of Infectious Diseases & Hospital Epidemiology, University Hospital Basel and University of Basel, 4031 Basel, Switzerland

**Keywords:** *Mycobacterium chimaera*, prosthetic-valve endocarditis, healthcare-associated infection

## Abstract

Background: Disseminated *Mycobacterium chimaera* (*M*. *chimaera*) infection following cardiac surgery has been associated with a high mortality. The long-term impact of surgery and the appropriate surgical approach are still matters of debate. Methods: From 2015 to 2019, seven patients with *M. chimaera* infection following cardiac surgery were isolated. Results: The median incubation time was 30 months (IQR 18–38). Echocardiography was unremarkable in three patients (43%). We decided to redo cardiac surgery in all patients and explanted all previously implanted prosthetic material. All explant cultures yielded *M. chimaera.* One patient (14%) died in-hospital seven months after the redo surgery. After a median follow-up of 59.6 months (IQR 39.1–69.6), we observed three infection relapses among the survivors (43%), presumably due to concomitant extracardiac infection and recurrent cardiac implant infection. Conclusions: *M. chimaera* infection following cardiac surgery is associated with a delayed and unspecific clinical presentation. Echocardiogaphy has a limited sensitivity for prosthetic valve infection with *M. chimaera*, and negative findings should not preclude the surgical decision. The extraction of all previously implanted material is crucial to achieving the source control, as the re-implantation of prosthetic material as well as uncontrolled extracardiac infection at the time of the redo cardiac surgery appear to be key factors for persisting/relapsing infection.

## 1. Introduction

Mycobacterium *chimaera* (*M. chimaera*) is an environmental, slowly growing, non-tuberculous mycobacterium first described in 2004 as a variant species of the *Mycobacterium avium*-complex (MAC) [1]. The first case of *M. chimaera* infection following cardiac surgery was described in 2013 in Zurich, Switzerland [2]. In the following years, over a hundred cases have been reported worldwide [3]. Water tanks of the heater-cooler device (HCD) Stockert 3T (LivaNova, London, UK), used for cardiopulmonary bypasses (CPB), have been identified as the point source of infection. The postulated mechanism was the airborne transmission of contaminated aerosols with *M. chimaera* from the HCD to the operating field during surgery [4]. Infections have been mostly described after prosthetic valve, aortic vascular graft and left-ventricular assist device implantation [5,6,7]. Infections limited to the sternum have been reported after a cardiac transplant and coronary bypass surgery [8,9]. The estimated risk of infection in recent years was 0.14/1000 procedures in the United Kingdom, 0.78/1000 procedures in Switzerland and 1/1000 to 1/10000 procedures in North America.

*M. chimaera* is an invasive germ potentially causing disseminated disease even in immunocompetent patients [6,10]. When inoculated with implanted material during cardiac surgery, the incubation period, as defined from the index surgery to the onset of symptoms, ranges from three months to twelve years [11,12]. Patients typically exhibit unspecific systemic symptoms such as fever, night sweats and weight loss as well as immunologic and multi-organ embolic manifestations [13]. *M. chimaera* is best identified using 16S rRNA sequencing on positive blood cultures or tissue samples [14]. After isolation, the extended diagnostic work-up of both cardiac and extracardiac manifestations is warranted. Transesophageal echocardiography (TEE) may detect vegetations, abscesses and valvular or prosthesis dysfunction. Positron emission tomography-computed tomography (PET-CT) with fluorine-18-fluorodeoxy-D-glucose (FDG) detects vascular graft infections and disseminated lesions and reinforces the suspicion of endocarditis in cases of inconclusive echocardiography. Chorioretinitis should be explored with fundoscopy. The recommended antimicrobial regimen is macrolide combined with ethambutol and rifamycin for disseminated MAC-infection in HIV-patients [14]. Moxifloxacin, clofazimine, bedaquiline, and linezolid have also been used for resistant strains and severe refractory disease or as alternative agents for patients with a poor tolerance to first-line therapy. From the time of diagnosis, six to twelve weeks of adjuvant therapy with amikacin is recommended [15,16]. However, due to the ability of *M. chimaera* to form biofilms on vascular and cardiac implants, the efficacy of antimicrobials is limited. Indeed, the medical management of *M. chimaera* infections alone following cardiac surgery has been associated with either no improvement or infection relapse [6,8,9]. Outcomes are poor, and the reported mortality ranges from 20% to 67% [14]. The benefit of the surgical removal of infected material must be weighed against the risk of reoperation, which is particularly high in patients with disseminated mycobacteriosis. Furthermore, the echocardiographic criteria of endocarditis may be lacking, even with the disseminated disease with *M. chimaera*, making the decision of redoing cardiac surgery challenging [8,9,17]. As for the surgical approach, many points have yet to be clarified. In 2020, an expert consensus advocated for the removal of all cardiac foreign material [14]. The ideal timing of surgery after the initiation of antimicrobials remains unclear. The best substitute after the removal of prosthetic valves or vascular grafts is still to be defined. Long-term follow-up data are lacking in clarifying whether the long-term benefit outweighs the immediate mortality risk of redo surgery.

To improve the knowledge of possible therapeutic strategies, we reported our experience with a long-term follow-up after the surgical management of *M*. *chimaera* infection.

## 2. Methods

### 2.1. Patients’ Selection, Diagnosis and the Initiation of Antimicrobial Therapy

From 2015 to 2019 at the University Hospital of Basel, we identified seven cases of *M. chimaera* infections following cardiac operations performed between 2013 and 2014 in our center. All patients had prosthetic material implanted during the index surgery. Individual patients’ characteristics at the index surgery are listed in Table 1. In Case n°4, the patient had a permanent dual-chamber pacemaker implanted six months after surgery due to high-grade atrioventricular block (AVB).

The diagnosis was established on positive blood or tissue cultures for mycobacteria and further species identification with either genus-specific PCR or 16S-rRNA sequencing. Resistance testing was performed at the Swiss National Reference Center for Mycobacteria (Zurich, Switzerland). The resistance testing of all prescribed antimicrobials was assessed for all isolates. The resistance tests were interpreted for clarithromycin, amikacin, moxifloxacin and linezolid according to the Clinical and Laboratory Standards Institute breakpoints.

After establishing the diagnosis and initiating antimicrobial therapy, all cases were evaluated in a multidisciplinary team including infectious disease specialists, cardiologists, anesthesiologists and cardiac surgeons. First, follow-up blood cultures were obtained three to four weeks after the initiation of antibiotic therapy. We decided on redoing cardiac surgery for all patients to explant the presumably infected material. All patients underwent redo cardiac surgery with negative blood cultures.

### 2.2. Follow-Up after Redo Surgery

The antimicrobial regimen was unchanged after the redo cardiac surgery. After hospital discharge, the patients were monitored by the division of infectious disease on a monthly basis and later on with delayed controls depending on the individual evolution. For patients with suspected relapsing infection or positive follow-up blood cultures, PET-CT, echocardiography and symptoms-oriented imaging were performed with a low threshold.

### 2.3. Healthcare Measures

As the first case was identified, we analyzed water and aerosols from our two Stockert 3T HCD units. *M. chimaera* was isolated from both samples. Based on the recommendations of the Swiss Chimaera Task Force, we replaced the two units and built a dedicated space outside the operating room to store and control the HCD remotely [18]. We also established a strict weekly chlorine-based disinfection protocol of the HCD water circuits, according to the manufacturer’s instructions, as well as a monthly analysis for the atypical mycobacteria of HCD water and air samples.

## 3. Results

### 3.1. Frequency of Disease

During the observation period, the incidence of cardiac-surgery-associated *M. chimaera* infection was 1.4 patients/year. Accounting for the number of cardiac procedures with CPB at the University Hospital of Basel during this period (*n* = 2817), the cumulative risk of *M. chimaera* infection was approximately 2.5/1000 procedures.

Notably, since the implementation of the above-mentioned healthcare measures, no further infections with *M. chimaera* have been detected in our center.

### 3.2. Patients’ Presentation and Paraclinical Findings

The median incubation time, as defined from the index surgery to the onset of symptoms, was 30 months (IQR 18–38). The diagnosis was often delayed due to the unspecific clinical presentation. Of five cases with available laboratory tests at the time of diagnosis, only one had markedly elevated C-reactive protein levels. The median time between the onset of symptoms and the diagnosis was 5.0 months (3.2–9.5). Sarcoidosis was misdiagnosed in two cases (28%), which was responsible for the longest delay (17 months).

Echocardiography was unremarkable in three patients (43%). In a single case, echocardiography atypically showed an aortic root enlargement. The most common echocardiographic finding was an aortic root abscess (43%). In three cases (43%), PET-CT revealed a marked FDG enhancement along the implanted cardiac device (see Figure 1). First, negative blood cultures after the initiation of antimicrobials were observed after a median duration of 4 weeks (IQR 4–5 weeks). Table 2 summarizes the patients’ clinical and paraclinical findings at presentation.

### 3.3. Surgical Management

Redo surgery was performed in all cases after a median period of 3.6 months (2.7–4.2) following diagnosis. The cannulation strategy was left to the discretion of the surgeon. An extracorporeal cytokine adsorber (CytoSorb^®^, CytoSorbents Corporation, Monmouth Junction (NJ), USA) was used in five cases (86%).

The most common operative finding was a large, caseous periannular abscess (43%), followed by vegetations on the prosthetic valve (28%) (see Figure 2 and Figure 3). In **Case n°3**, the inspection of the mitral valve as well as of the explanted ring was unremarkable.

Our surgical concept was to explant all the prosthetic material. We opted for cryopreserved aortic homografts for the replacement of aortic valve bioprostheses, aortic bioprosthetic conduits and aortic interposition grafts, regardless of the extent of the native valve or the graft destruction (see Figure 4).

In Case n°3, the infected annuloplasty ring and neo-chordae were replaced with a new annuloplasty ring and a single neo-chorda.

In Case n°1, persistent third-degree AV block prompted the implantation of a dual-chamber permanent epicardial pacemaker. In Case n°4, the previously implanted pacemaker leads were replaced with epicardial leads. The generator was also replaced during redo surgery.

### 3.4. Postoperative Outcomes and Infection Control

One patient (14%) died in-hospital seven months after the redo surgery. After several early surgical revisions due to persistent postoperative bleeding, the patient experienced numerous complications including anuric renal failure, pneumonia with respiratory failure and gastrointestinal bleeding with hemorrhagic shock. He finally died from sepsis caused by multiresistent Pseudomonas aeruginosa. The remaining six patients were alive after a median follow-up of 59.6 months (39.1–69.6).

We observed a single case (14%) of early resistance/elevation of the minimum inhibitory concentration of M. chimaera with ethambutol, in which we substituted the agent with bedaquiline (Case n°7).

#### Three Patients (43%) Experienced Infection Relapse

In Case n°2, eight months after the redo surgery, the patient presented with *M. chimaera* infection relapse affecting a right hip prosthesis and the lumbar spine. The infection was controlled with the removal of the hip prosthesis and the surgical debridement of the spine. Retrospectively, PET-CT images prior to the redo cardiac surgery were suggestive of hip prosthesis infection. No progression or relapse of infection was noted under suppressive antimicrobial therapy, and a hip prosthesis was re-implanted after twelve months.

In Case n°3, six months after the redo surgery, the patient presented with disseminated granulomatous manifestations and blood cultures attesting *M. chimaera*. PET-CT demonstrated moderate FDG uptake along the second mitral ring, and TEE was unremarkable. Due to the uncontrolled infection despite the antimicrobials, we opted for surgical exploration anew, the extraction of the mitral ring and mitral valve reconstruction with the transfer of the secondary autologous chordae. The mitral valve after the second redo showed moderate regurgitation. The ring cultures attested *M. chimaera.* PET-CT, fourteen months after this second reoperation, was not suggestive of infection persistence or relapse under suppressive therapy.

In Case n°7, ten months after the redo surgery, the patient presented with disseminated *M. chimaera* re-infection involving the right sternoclavicular joint, sternum and lumbar spine. The retrospective analysis of PET-CT images prior to the redo surgery revealed the involvement of the right sternoclavicular joint. We decided on the surgical debridement of all affected sites including partial sternectomy and subsequent reconstruction with a pectoralis advancement flap. Shortly after discharge, the patient was readmitted due to hemorrhagic shock due to a tear in the aortic homograft, presumably caused by the adjacent remnant sternal border. We performed an emergent replacement of the distal homograft on a cardiopulmonary bypass using a bovine pericardial tube. The cultures of the explanted homograft showed no growth of *M. chimaera*. Histopathology showed granulomatous inflammation compatible with mycobacterial infection. After discharge, the infection persisted in the lumbar spine, requiring local surgical debridement and reinforcement of the antimicrobial therapy with clofazimine. Table 3 summarizes the patients’ surgical characteristics at the redo cardiac surgery and their postoperative outcomes. Table 4 provides individual information regarding antimicrobial therapy management and infection control. The detailed individual surgical management and outcomes are listed in Appendix A.

## 4. Discussion

### 4.1. Clinical Presentation, Diagnosis and Surgical Management

The patients’ presentation was unspecific. Sarcoidosis was misdiagnosed in two cases. The systemic, granulomatous-like features of *M. chimaera* infections are known to prompt the misdiagnosis of sarcoidosis [2,19].

We decided on redo surgery and the explant of all cardiac foreign material in all cases. Importantly, surgery was decided upon in the absence of the echocardiographic criteria of endocarditis in three patients. In one case, echocardiography was very atypical (aortic root enlargement). *M. chimaera* was isolated in all explant cultures. This observation is consistent with the literature, which reports an inconstant TEE sensitivity for *M. chimaera* infections, ranging from 38% to 83% [9,13,17,19,20]. Additionally, cases with normal initial echocardiography have been further diagnosed with *M. chimaera* endocarditis by either follow-up TEE or post-mortem examination [8]. The isolation of *M. chimaera* in all cardiac explant cultures in our series reinforces the hypothesis that the implanted prosthetic material constitutes the nidus of infection, whose removal is mandatory for infection control. Therefore, inconclusive echocardiography should not preclude the decision of redo surgery. PET-CT should also be performed with a low threshold as a complement to echocardiography and to assess the extracardiac extent of the disease.

### 4.2. Surgical Approach and Immediate Postoperative Outcomes

Except for one case, we sought to avoid re-implanting prosthetic material and opted for biological substitutes. We selected cryopreserved aortic homografts for the replacement of aortic valve bioprostheses, bioprosthetic valve conduits and aortic polyester grafts. Although controversial, previously published data suggest lower rates of re-infection using homografts compared to those with mechanical or biological prostheses for prosthetic aortic valve endocarditis [21,22,23,24,25,26]. The durability of cryopreserved homografts is acceptable, with the reported freedom from reoperation due to structural valve deterioration as high as 89% at 15 years [27]. However, durability was proven to be related to the patient’s age at implantation, and concerns are raised about homografts in young patients [28].

In Case n°3, considering that mitral valve repair without an annuloplasty ring would lead to the exposure to significant residual mitral regurgitation, we initially opted for ring- and Gore-Tex chordae replacement. As the patient presented with disseminated infection again six months later, we opted for the surgical extraction of the ring and a prosthesis-free repair at the cost of moderate residual mitral regurgitation.

The use of a cytokine adsorber was shown to reduce the vasopressor needs after cardiac surgery in patients with native valve endocarditis [29]. Whether the outcomes have been positively influenced using Cytosorb® in our patients remains hypothetical. Recent publications seriously called into question the benefit of cytokine adsorbers in cardiac surgery, since a reduction in neither the pro-inflammatory response nor mortality has been demonstrated [30,31]. In a retrospective study on cardiac surgery for infective endocarditis at the University Hospital of Basel, Santer et al. even reported longer hospital stays and more frequent surgical revisions for postoperative bleeding with the use of a cytokine adsorber [32].

As for the pacemakers, those implanted prior to the redo surgery were removed and replaced with epicardial systems to avoid intravascular prosthetic material. Patients with new AVB at the time of the redo were also implanted with epicardial leads.

We frequently observed postoperative bleeding requiring surgical revisions (57%). Except for one case, we dealt with diffuse bleeding rather than surgical bleeding. This finding is explainable by the coagulopathy induced by the long CPB times inherent in these complex reoperations (see Table 2).

### 4.3. Long-Term Outcomes and Infection Control

Long-term, multiple-antimicrobial therapy was challenging due to the numerous adverse reactions. However, we strove to maintain antibiotic regimens in line with the current recommendations [14]. Infection relapses occurred after varying postoperative time intervals and with diverse clinical presentations. We observed a single case of the resistance of *M. chimaera* with ethambutol, and this case was associated with relapsing/persisting infection (Case n°7). However, several factors potentially played a role in the infection relapse in this case (see below).

In Case n°2, undiagnosed concomitant hip prosthesis infection presumably led to the secondary dissemination of *M. chimaera* to the lumbar spine eight months after the redo surgery. This mode of relapse illustrates the inability of antibiotics to sterilize *M. chimaera*-infected prosthetic material. Therefore, this patient may have benefited from a prompter surgical extraction of his hip prosthesis.

In Case n°3, cross contamination of the second ring presumably occurred after the redo surgery despite four months of prior antimicrobial therapy. This case was notable for the infection relapse after the cardiac re-implantation of prosthetic material during the redo surgery. This observation definitely supports the avoidance of new prosthetic material until the eradication of *M. chimaera* is achieved. However, the infect-free duration corresponding to the eradication of *M. chimaera* has not yet been defined. In our series, we discontinued antimicrobials in three patients with controlled infection after two years, with no evidence of relapse to date.

In Case n°7, several scenarios are to be considered to explain the infection relapse. As mentioned above, a resistance to ethambutol was noted early after the antimicrobial therapy initiation, and the agent was promptly substituted with bedaquiline. In addition, the infection of the right sternoclavicular joint may have spread locally to the right clavicle and sternum. Alternatively, concomitant spondylodiscitis, possibly masked by the initial use of steroids due to falsely diagnosed sarcoidosis, may have spread hematogenously. Overall, both this case and Case n°2 (hip prosthesis) emphasize the importance of excluding extracardiac infection sites prior to redoing cardiac surgery.

In Case n°7, the patient’s medical history was also relevant for splenectomy due to hereditary spherocytosis. The role of splenectomy as a risk factor for infection relapse remains hypothetical. If the literature mentions rare overwhelming post-splenectomy infections with *M. tuberculosis*, evidence is lacking regarding the role of impaired phagocytosis associated with splenectomy in non-tuberculous mycobacterial infections [33,34].

In recent publications, the mortality associated with *M. chimaera* infection following cardiac surgery ranged from 20 to 60% [4,6,8,9,13,17,35]. These publications included patients managed both conservatively and surgically. In the long term, limited data are available, so the long-term impact of redo surgery on survival remains unclear. In 2020, Julian et al. published their multi-center experience involving 28 patients with *M. chimaera* infections [9]. After 3 to 7 years of follow-up, the authors reported 56% mortality including postmortem diagnoses. Importantly, survivors after cardiac prosthesis removal/replacement had their infection more frequently under control than patients treated conservatively (67% versus 25%, *p* > 0.14). Although our cohort was more homogenous and not of a comparable size, we observed 14% mortality after a median follow-up of barely 5 years. Altogether, these observations support an aggressive approach with the systematic surgical removal of prosthetic material to achieve the source control in *M. chimaera* infection following cardiac surgery.

## 5. Limitations

The small number of cases limits the interpretation based on this report. Furthermore, we report a single-center experience. The lack of a control group with conservative management is a theoretical methodological flaw and limits the statistical inference of our observations. Concerning mitral valve endocarditis with *M. chimaera*, our experience is limited to a single case (Case n°3). More studies with a larger number of patients and a detailed surgical approach would be necessary to better define the role of surgery as well as the ideal substitute in *M. chimaera* infections following cardiac surgery. However, since the identification of the point-source and the implementation of healthcare measures have effectively prevented further infections of *M. chimaera* following cardiac surgery, the number of cases is decreasing worldwide. This salutary evolution will make it more difficult to establish evidence in the future and emphasizes the importance of reporting identified cases.

## 6. Conclusions

*M. chimaera* infection following cardiac surgery is associated with a delayed and unspecific clinical presentation. Echocardiography has a limited sensitivity for detecting prosthetic-valve infection with *M. chimaera*, and negative findings should not preclude the surgical decision. The extraction of all prior implanted material is crucial to achieving the source control, as the re-implantation of prosthetic material at the time of redo surgery as well as uncontrolled extracardiac infection appear to be key factors for persisting/relapsing infection.

## Figures and Tables

**Figure 1 jcm-12-00948-f001:**
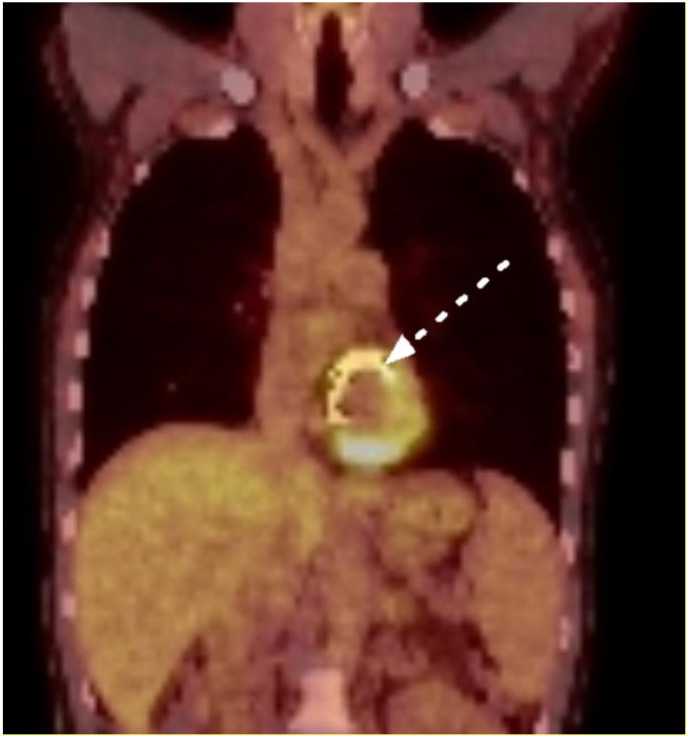
Positron-emission tomography: marked enhancement along the mitral annuloplasty ring (white arrow, Case n°4).

**Figure 2 jcm-12-00948-f002:**
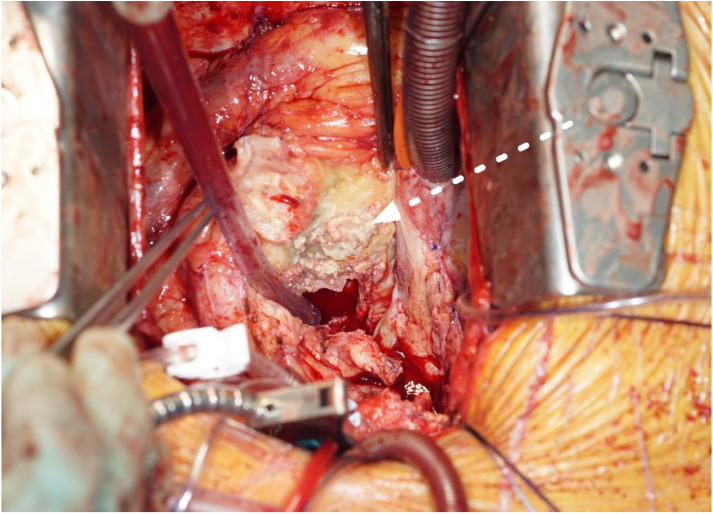
Aortic periannular abscess extending to the right atrial roof (white arrow, cranial view, Case n°6).

**Figure 3 jcm-12-00948-f003:**
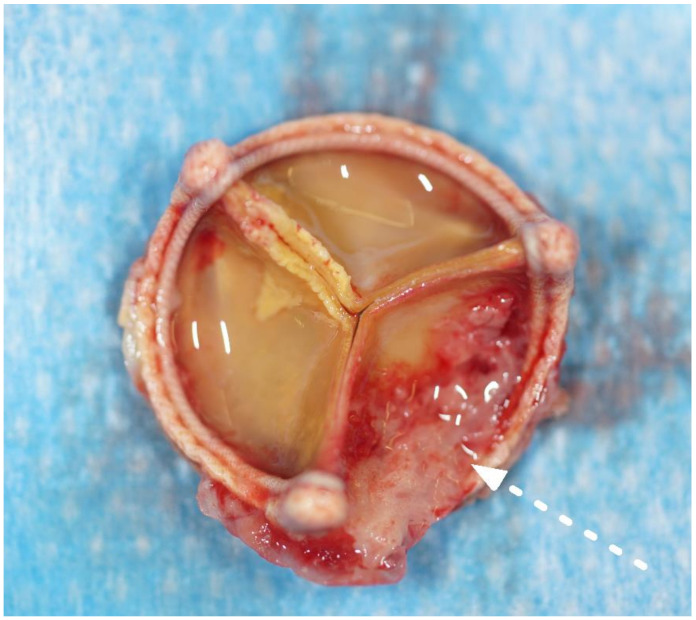
Explanted aortic valve bioprosthesis with fibrous, granulomatous deposits on the aortic size of the leaflets (white arrow, Case n°2).

**Figure 4 jcm-12-00948-f004:**
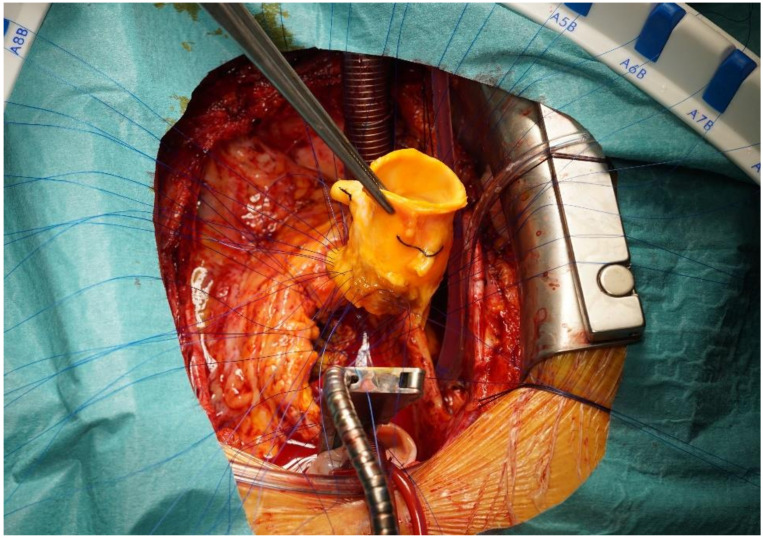
Aortic homograft implantation using polypropylene interrupted sutures.

**Table 1 jcm-12-00948-t001:** Patients’ characteristics at the index surgery.

Case n°	Age	Gender	Index Diagnosis	Index Operation	Implant
1	62	male	Aortic valve endocarditis (S. *dysgalactiae*)	Aortic valve replacement	Stented bovine bioprosthesis
2	63	male	Aortic valve regurgitation Ascending aortic aneurysm	Aortic root replacement	Bioprosthetic valve conduit (stented bovine valve)
3	48	male	Mitral valve prolapse	Mitral valve reconstruction	Annuloplasty ring Gore-Tex neochorda
4	55	male	Aortic root aneurysm Single-vessel CAD	Aortic root re-implantationSingle CABG	Polyester interposition graft
5	63	male	Aortic valve stenosis	Aortic valve replacement	Stented bovine bioprosthesis
6	71	male	Aortic valve regurgitation Ascending aortic aneurysm	Aortic root replacement	Bioprosthetic valve conduit (stented bovine valve)
7	48	male	Ascending aortic aneurysm	Ascending aortic and hemiarch replacement	Polyester interposition graft

CAD: coronary artery disease; CABG: coronary artery bypass graft.

**Table 2 jcm-12-00948-t002:** Patients’ clinical and paraclinical findings.

Variable	Median (IQR)/ Number of Patients (Percentage)
Median incubation time (months) ^ⴕ^	30 (18–38)
Clinical symptoms	
Fatigue	4 (57)
Weight loss	4 (57)
Night sweat	1 (14)
Fever	4 (57)
Dyspnea	1 (14)
Asymptomatic	1 (14)
Laboratory tests	
C-reactive protein (mg/L)	9.4 (1.9–70.5)
TEE findings	
Aortic root abscess	3 (43)
Severe aortic valve regurgitation	1 (14)
Aortic root enlargement	1 (14)
Unremarkable	3 (43)
PET-CT cardiac findings	
Strongly positive	3 (43)
Slightly positive	1 (14)
Unremarkable	2 (28)
Extracardiac involvement	
Chorioretinitis	5 (71)
Nephritis	3 (43)
Myelitis	2 (28)
Hepatitis	5 (71)
Cerebritis	3 (43)
Pneumonitis	1 (14)
Median time to diagnosis (months)	5.0 (3.2–9.5)

^ⴕ^ defined from the index surgery to the onset of symptoms; IQR: interquartile range; TEE: transesophageal echocardiography; PET-CT: Positron emission tomography-computed tomography.

**Table 3 jcm-12-00948-t003:** Patients’ surgical characteristics at the first redo cardiac surgery and postoperative outcomes.

Variable	Median (IQR)/ Number of Patients (Percentage)
Time from diagnosis to redo surgery (months)	3.6 (2.7–4.2)
Substitute	
Aortic homograft	6 (86)
Bovine pericardial tube	2 (28)
Annuloplasty ring and Gore-Tex neo-chordae	1 (14)
CPB cannulation site	
Aorta	5 (71)
Femoral artery	1 (14)
Right axillary artery	1 (14)
Median CPB time (min)	161 (149–168)
Median aortic cross-clamp time (min)	141 (128–182)
Postoperative bleeding	4 (57)
Positive explant cultures	7 (100)
Median hospital length of stay (days)	16 (11–24)
Antibiotic resistance	1 (14)
Infection relapse	3 (43)
Time to infection relapse (months)	8 (7–9)
Second redo cardiac surgery	2 (28)
Follow-up (months)	60 (39–70)
Status	
Alive	6 (86)
Dead	1 (14)

IQR: interquartile range; CPB: cardiopulmonary bypass.

**Table 4 jcm-12-00948-t004:** Antimicrobial therapy management and infection control.

Case n°	Initial AB Regimen	Duration of AB Therapy (Months)	Side Effects	AB Cessation	Reason for AB Cessation	Alternative AB Agents	Duration of Alternative AB Agents (Months)	Infection Relapse/Persistence	Time to Relapse (Months)	FU Duration (Months)	Status
1	Clarithromycin	38		Yes	Clinical cure			No		69	Alive
	Rifabutin		Pancytopenia								
	Moxifloxacin	1	Type IV allergy		Side effects						
	Ethambutol	20	Visual disturbances								
	Amikacin	4	Hearing lossVertigo								
2	Clarithromycin	61		Yes	Treatment simplification	Azithromycin	12	Yes	8	73	Alive
	Rifabutin	60			Stable disease						
	Moxifloxacin	48				Clofazimine	24				
	Ethambutol	73		No							
3	Clarithromycin	68		Yes	Clinical cure			Yes	26	70	Alive
	Rifabutin	48	Hepatitis	Intermittent	Side effects	Amikacin	6				
	Moxifloxacin	9	Arthralgia, Tendinopathy	Yes	Side effects	Bedaquiline	19				
	Ethambutol	68			Clinical cure						
4	Clarithromycin	48		No				No		49	Alive
	Moxifloxacin										
	Rifabutin	1	Nausea	Yes	Side effects						
	Ethambutol	48		No							
	Amikacin	4		Yes	End of treatment						
5	Clarithromycin	25		Yes	Clinical cure			No		60	Alive
	Rifabutin										
	Moxifloxacin		*C. difficile* Colitis								
	Ethambutol										
	Amikacin	4			End of treatment						
6	Clarithromycin	7	Pancytopenia	Yes	Side effects			No		7	Deceased
	Rifabutin		Leukopenia								
	Moxifloxacin		Pancytopenia								
	Amikacin		Hearing loss								
	Ethambutol		Leukopenia								
7	Clarithromycin	34		No				Yes	10	34	Alive
	Rifampicin										
	Moxifloxacin		Polyneuropathy	Yes	Side effects	Clofazimine	1				
	Amikacin	2	Hearing loss			Bedaquiline	28				
	Ethambutol	1			Resistance						

AB: antibiotic; FU: follow-up.

## Data Availability

The data presented in this study are available on reasonable request from the corresponding author.

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
