# Peer review of "Long-Term Follow-Up after Mycobacterium Chimaera Infection Following Cardiac Surgery: Single-Center Experience"

_jcm, 2023, doi:10.3390/jcm12030948_

Round 1

Reviewer 1 Report

Dear authors, thank you for your interesting manuscript. I would like to encourage you to comment on the following points:

Please specifiy the us of antiinfectiva and antibiotics for each case including resistancy, and duration of therapy.

Please comment on laboratory markers with CRP, PCT, e.g. for the cases

What are your suggested strategies to avoid such infections?

Author Response

Reviewer #1

Comment:

Please specify the use of anti-infective and antibiotics for each case including resistance, and duration of therapy.

Answer:

As suggested by the academic Editor and by Reviewer#1 and #3, we added information on individual antimicrobial therapy management and infection control. This information was listed in Table 4. For more clarity, Table 1 was placed in the Methods section.

We observed a single case (14%) of early resistance / elevation of the minimum inhibitory concentration with ethambutol, in which the agent was substituted with bedaquiline (Case n°7).

Modification:

l.86, Methods: Table 1 was placed in the Methods sections

l.254, Results: Table 3 was created.

l.209, Results: We observed a single case (14%) of early resistance / elevation of the minimum inhibitory concentration of M. chimaera with ethambutol, in which we substituted the agent with bedaquiline (Case n°7).

Comment:

Please comment on laboratory markers with CRP, PCT, e.g. for the cases

Answer:

# Due to external initial work-up for two patients, biomarkers were not available for all cases. At presentation, we noted a single case with markedly elevated (125mg/l) and four cases with normal or slightly elevated (<15mg/l) CRP levels. PCT values were not available.

Modification:

  1. 131, Results: Of five cases with available laboratory tests at the time of diagnosis, only one had markedly elevated C-reactive protein levels.

Table 2: Laboratory tests were included

Comment:

What are your suggested strategies to avoid such infections?

Answer:

Health-care strategies to avoid such infection have been specified in the Methods section: health-care measures. Moreover, we implemented monthly analyses of water and air samples from the Heater-cooler device.

Of note, since the implementation of the above mentioned health-care measures, no further infections with M. chimaera have been detected in our center.

Modification:

  1. 115, Methods: We also established […] monthly analysis for atypical mycobacteria of HCD water and air samples.

l.125, Results: Of note, since the implementation of the above mentioned health-care measures, no further infections with M. chimaera have been detected in our center.

Reviewer 2 Report

Schaeffer, Kuster, and colleagues report their findings after reoperation for Mycobacterium chimaera-associated endocarditis in their article "Long-term follow-up after Mycobycterium chimaera infection following cardiac surgery: single-center experience."

The authors present seven cases with postoperative evidence of M. chimaera after previous cardiac surgery. The cases are well presented, yet a meaningful conclusion is lacking. As presented, the article is more of a case series than an original article.

1. Were there any cases treated conservatively during the 2015-2019 period? And if so, where were there differences in outcome?

2. were there repeat PET-CT/blood cultures after a reasonable time after antibiotic treatment? If so, were these still positive?

3. What was the final decision for surgical therapy, especially in patients with unremarkable PET-CT findings?

Thanks to the authors for the article.

Author Response

Reviewer #2

Comment:

Schaeffer, Kuster, and colleagues report their findings after reoperation for Mycobacterium chimaera-associated endocarditis in their article "Long-term follow-up after Mycobycterium chimaera infection following cardiac surgery: single-center experience."

The authors present seven cases with postoperative evidence of M. chimaera after previous cardiac surgery. The cases are well presented, yet a meaningful conclusion is lacking. As presented, the article is more of a case series than an original article.

Comment:

Were there any cases treated conservatively during the 2015-2019 period? And if so, where were there differences in outcome?

Answer:

There were no cases treated conservatively during the study period in our center.

As mentioned in the Introduction section, conservative management of disseminated M. chimaera infection after cardiac surgery has been associated with high mortality. Failure of medical therapy alone has been attributed to the limited efficacy of antimicrobials on implant-associated infections, due the ability of mycobacteria to form protective biofilms. Based on these premises, we opted for an aggressive infection management with surgical extraction of prosthetic material in all cases.

Nonetheless, the lack of control group with conservative management is a theoretical methodological flaw in our study and limits the statistical inference of our observations.

As suggested by Reviewer#2, we mentioned this point in the Limitations section.

Modification:

l.356, Limitations: The lack of control group with conservative management is a theoretical methodological flaw and limits the statistical inference of our observations.

Comment:

Were there repeat PET-CT/blood cultures after a reasonable time after antibiotic treatment? If so, were these still positive?

Answer:

Blood cultures were routinely obtained 3 to 4 weeks after initiation of antibiotic therapy. First negative blood cultures after treatment initiation were observed after a median duration of 4 weeks (IQR 4-5 weeks). All patients underwent redo cardiac surgery with negative blood cultures. For patients with suspected relapsing infection or positive follow-up blood cultures, PET-CT, echocardiography, and symptoms-oriented imaging were performed with low threshold.

As suggested by Reviewer#2, we specified the follow-up management and the time to first negative blood cultures after initiation if the antimicrobial therapy.

Modification:

  1. 98, Methods: First follow-up blood cultures were obtained three to four weeks after initiation of antibiotic therapy.
  2. 100, Methods: All patients underwent redo cardiac surgery with negative blood cultures.
  3. 105, Methods: For patients with suspected relapsing infection or positive follow-up blood cultures, PET-CT, echocardiography, and symptoms-oriented imaging were performed with low threshold.
  4. 135, Results: First negative blood cultures after initiation of antimicrobials were observed after a median duration of 4 weeks (IQR 4-5 weeks).

Comment:

What was the final decision for surgical therapy, especially in patients with unremarkable PET-CT findings?

Answer:

PET-CT was routinely used at the time of diagnosis for cases with inconclusive TEE and for detection of extracardiac involvement. However, for two cases, both TEE and PET-CT were indeed unremarkable.

As mentioned in the Discussion section, our hypothesis, which we share with others, was that the implanted prosthetic material constitutes the nidus of infection, whose removal is mandatory for infection control (Scriven et al. Clin Microbiol Infect. 2018). Therefore, we decided on surgical redo for all cases regardless of echocardiographic and PET-CT findings.

Modification:

None.

Reviewer 3 Report

The authors describe their experience with 7 patients who had mycobacterium chimaera infection after cardiac surgery. All patients had redo surgery and relapse was 47%. The paper nicely summarizes the current information on this infection and provides some useful strategies in dealing with this infection, such as obtaining PET and recommendation of removing all prosthetic material even in the absence of TEE findings. I have a few comments.

Can authors provide the total number of cases during this time or in another word the “incidence” of this infection in the timeframe of the study, please?

Could you please clearly indicate what was the antibiotic regimen after surgery? Were all these 7 patients placed on suppressive antimicrobial therapy and for how long. This was not very clear from the paper.

Infection of a mitral annuloplasty ring is very uncommon. And having that recur in patient #3 is quite surprising to me. Can authors describe the operative findings during this case and was there any mitral annular calcification or abscess or extra cardiac infection that could explain the recurrence of the infection.

In table 3: there seems to be a typo. The median CPB time is 1 min! and median circulatory arrest time is 141 min. Could you please correct this if necessary.

In case #2 was did you have a PET scan during the initial diagnosis? Was there any evidence of hip involvement on that?

In conclusion section, I suggest authors refrain from their “advise” to use biological material (homograft) instead of mechanical or tissue prosthesis. As they mentioned earlier in the paper, this is an area of controversy and since the authors don’t have any evidence for this, they should refrain from any strong recommendation.

 Thank you for the opportunity to review your work. 

Author Response

Reviewer #3

The authors describe their experience with 7 patients who had mycobacterium chimaera infection after cardiac surgery. All patients had redo surgery and relapse was 47%. The paper nicely summarizes the current information on this infection and provides some useful strategies in dealing with this infection, such as obtaining PET and recommendation of removing all prosthetic material even in the absence of TEE findings. I have a few comments.

Comment:

Can authors provide the total number of cases during this time or in another word the “incidence” of this infection in the timeframe of the study, please?

Answer:

Dear Reviewer#3,

Thank you for your nice comment.

As suggested by Reviewer#3, we added additional information on the reported incidence for several western countries in recent years, as well as the cumulative risk of disease in our center during the study period.

Modification:

  1. 40, Introduction: The estimated risk of infection in recent years was 0.14/1000 procedures in the United Kingdom, 0.78/1000 procedures in Switzerland, and 1/1000 to 1/10000 procedures in North America.

l.121, Results: “Frequency of disease”

During the observation period (5 years), the incidence of cardiac-surgery associated M. chimaera infection was 1.4 patients / year. Accounting for the number cardiac procedures with CPB at the University Hospital of Basel during this period (n=2817), the cumulative risk of M. chimaera infection was approximately 2.5 / 1000 procedures.

Comment:

Could you please clearly indicate what was the antibiotic regimen after surgery? Were all these 7 patients placed on suppressive antimicrobial therapy and for how long. This was not very clear from the paper.

Answer:

As suggested by the academic Editor and by Reviewer#1 and #3, we added information on individual antimicrobial therapy management and infection control. This information was listed in Table 4. For more clarity, Table 1 was placed in the Methods section.

The antimicrobial regimen was unchanged after redo cardiac surgery. Two patients were placed on suppressive antimicrobial therapy due to persisting/relapsing infection. For these two cases, a lifelong therapy is foreseen.

Modification:

l.86, Methods: Table 1 is now in the Methods sections

l.103, Methods: The antimicrobial regimen was unchanged after redo cardiac surgery.

l.254, Results: Table 4 was created.

Comment:

Infection of a mitral annuloplasty ring is very uncommon. And having that recur in patient #3 is quite surprising to me. Can authors describe the operative findings during this case and was there any mitral annular calcification or abscess or extra cardiac infection that could explain the recurrence of the infection.

Answer:

Annuloplasty-ring associated endocarditis and its recurrence after ring replacement was also very surprising to us. The very low probability of this scenario explains why we did not perform a prosthetic-free mitral valve repair in the first place. The patient suffered Barlow disease and the inspection of the valve as well as of the explanted ring, on neither the first nor the second time, did reveal calcification or abscess. PET-CT enhancement prior to redo surgery was limited to the mitral ring.

As suggested by Reviewer#3, we provided additional descriptive information on Case n°3.

Modification:

  1. 171, Results: In Case n°3, the inspection of the mitral valve as well as of the explanted ring was unremarkable.

Comment:

In table 3: there seems to be a typo. The median CPB time is 1 min! and median circulatory arrest time is 141 min. Could you please correct this if necessary.

Answer:

Thank you for this comment. We accidentally inverted the values of CPB time and Cross-clamp time.

As pointed by Reviewer#3, we corrected the values in Table 3.

Modification:

l.251, Table 3:

Median CPB time (min) 161 (149 – 168)

Median aortic cross-clamp time (min) 141 (128 – 182)

Comment:

In case #2 was did you have a PET scan during the initial diagnosis? Was there any evidence of hip involvement on that?

Answer:

A retrospective analysis of the PET-CT images acquired prior to redo cardiac surgery yielded some evidence of a concomitant hip prosthesis infection on the right side. At the time of diagnosis, we prioritized the cardiac surgical intervention, as the point source of infection obviously was the heart. Twelve months later, the hip prosthesis was extracted after the routine PET-CT control demonstrated persisting enhancement of the right hip, and cultures from the hip puncture were positive. This mode of relapse illustrates here again the inability of antibiotics to sterilize M. chimaera infected prosthetic material. Therefore, this patient may have benefit from a prompter surgical extraction of his hip prosthesis.

As suggested by Reviewer#3, we specified preoperative PET-CT findings on Case n°2 in the Results section and commented this point in the Discussion section.

Modification:

l.217, Results: Retrospectively, PET-CT images prior to redo cardiac surgery were suggestive of hip prosthesis infection.

  1. 316, Discussion: This mode of relapse illustrates here again the inability of antibiotics to sterilize M. chimaera infected prosthetic material. Therefore, this patient may have benefit from a prompter surgical extraction of his hip prosthesis.

Comment:

In conclusion section, I suggest authors refrain from their “advise” to use biological material (homograft) instead of mechanical or tissue prosthesis. As they mentioned earlier in the paper, this is an area of controversy and since the authors don’t have any evidence for this, they should refrain from any strong recommendation.

Answer:

Aware of the limited efficacy of antibiotics on M. chimaera infected prosthetic material, and as illustrated by Case n°3 (mitral ring annuloplasty), our conviction remains that the first-line substitute for redo surgery in these cases should be biological. However, as mentioned by Reviewer#3, this is an area of controversy and strong evidence supporting this concept is lacking. However, it is questionable whether it will be ever possible to obtain evidence on this issue.

As suggested by Reviewer#3, we revised our conclusions.

Modification:

  1. 360, Conclusions (last sentence deleted)

Round 2

Reviewer 2 Report

The authors make a significant improvement in their articles. I have no further questions.

Thanks to the authors for the opportunity to review this work.